# Oversized Electrical Appliance Impacts on Condominium Energy Efficiency and Cost-Effectiveness Management: Experts' Perspectives

**Techatat Buranaaudsawakul and Kittipol Wisaeng ***

Mahasarakham Business School, Mahasarakham University, Mahasarakham 44150, Thailand; 62010990004@msu.ac.th
* Correspondence: kittipol.w@acc.msu.ac.th

**Abstract:** A direct use approach incorporating a cost approach assumed that replacing oversized electrical appliances with those better fit to actual energy consumption can reduce energy consumption, optimizing capacities of the new appliances to the maximum while reducing electricity costs. This study aimed to verify the assumption that the size of appliances has impacts on energy consumption and cost effectiveness. A mixed-method approach included these instruments for data elicitations (i.e., a questionnaire, data records of 485 transformers, two assessments of condominium technical caretakers, and two in-depth interviews of electrical engineering experts). The findings revealed that most condominiums installed electric appliances that are too large for their actual energy usage, which lies between 5.4% and 7.1% of the capacity. This study therefore proposed a total cost reduction of 54% by downsizing these appliances (i.e., MV Switchgear 2 sets, dry type transformer 2 sets 80,000, LV Cable 10 m. (XLPE), main distribution board, Busduct (MDB-DB), generator (20% of Tr.), and generator installation). Even though this analysis is limited to Bangkok, Thailand, this case may contribute decision-making on electrical appliance selection at early stage of investment or to downsize the currently installed appliances for the more energy efficient and cost-effective management of condominiums around the world.

**Keywords:** oversized electrical appliances; impacts on condominium; energy efficiency; cost-effectiveness

## 1. Introduction

Offices and residential buildings, including condominiums, in all countries around the world consume more energy and consumption is expected to grow. In terms of economic perspectives, this energy inefficiency has led to financial losses and other undesirable management consequences. Solutions that are economically viable for enhancing energy efficiency and effective electricity consumption management of buildings, have been proposed to energy efficiency management to enhance energy-saving buildings, especially for condominiums in big cities.

Demands for electricity usage in the condominiums in Bangkok, the capital of Thailand with an approximate population of 6 million residents, are expected to rise. Among the several factors of rising usage is energy inefficiency, resulting from disparities between electric appliances and actual energy consumption. The rising demands have called for investigations producing insights related to overall energy consumption to provide a guideline for the electricity consumption management of condominiums in Bangkok.

### 1.1. Prior Research on Offices and Residential Buildings

In recent decades, myriad attempts and interventions for offices, residential buildings and condominiums have been proposed. Some attempts have identified factors of high energy consumption. Others have drawn insights from lessons, case studies, and models

to further propose plans, designs, strategies, methods, measures, assessment, approaches, and solutions to enhance energy consumption management.

### 1.1.1. Factors of High Energy Consumption

Attempts to identify key factors of high energy consumption and barriers to energy efficiency management have been made. Key factors have been identified by various approaches and methods, including a multidimensional hybrid approach that combines energy monitoring and modeling to determine the factors of residential home energy consumption [1], urban heat on building energy use [2], and a comparison of the environmental performance of building properties using the Eco-Effect tool to determine the quality vs. energy efficiency impact [3]. Scholars have also proposed the drivers of observed changes over time of energy impacts such as the decomposition analysis of residential energy use in Hong Kong from 1990 to 2007 [4] and, later, in China from 2002 to 2010 [5].

### 1.1.2. Impacts on High Energy Consumption

The impacts of high energy consumption on economic and environmental development were also investigated. The investigations include coping with hikes in tariffs amid power shortages in urban households in Ethiopia [6], geothermal heat pumps interacting with low-temperature thermal microgrids and their benefits to the global and local environmental and energy [7], a value-based approach with a focus on creating sustainable alternatives in the built environment [8], developing a residential building's electricity consumption profile using smart meter data grouping [9], renewable energy conservation and the vast fields of the United States [10], combining power and gas in the transition to future urban energy systems [11], and sustainable home service to household services that promote ecological, social and economic sustainability [12].

### 1.1.3. Solutions at a Macro Level

Solutions to tackle solutions at a macro level have been provided. Some scholars have proposed models, such as a new dynamic simulation model for thermo-economic analysis and optimization of district heating systems [13], a business model which includes key issues and trends for the energy community [14], a transformation from the smart energy community into smart energy municipality [15] and key points to create further improvements for sustainable city strategies [16]. Similarly, implicit solutions in the form of lessons have also been provided, such as the lessons of transitioning to a low carbon society through energy communities in Brazil and Italy [17], sustainable energy action plans at a city level in Portugal [18] and shared photovoltaics for efficient, effective and fair allocation of costs and benefits in residential energy communities [19].

Other scholars have proposed deliberate designs, approaches, and methods. One study [20] combined various designs (including the integrated design, synthesis and the functioning of poly-generation systems, the implementation of an innovative mixed integer linear programming model, multi-purpose optimization for LCA emissions and costs) to improve energy efficiency. Similarly, Ref. [21] initiated a novel approach for managing the energy efficiency of office space HVAC systems through real-time big data analytics. In addition, another study [22] proposed a method for planning the operation for a home air conditioner, considering the nature of the installation environment by using a home energy management system (HEMS) to automatically control home appliances.

### 1.1.4. Solutions at a Micro Level

Several scholars have tackled the problems at a micro level. They have proposed strategies and applications. Among the strategies and applications are design strategies and quantification of energy flexibility in buildings in Italy [23], an application of variable speed drive (VSD) in the energy saving of electric motors [24], a techno-economic assessment of thermal and combined photovoltaic/fuel cell/battery energy system in Malaysian hospitals [25], a multi-stage and multi-objective optimization for additional energy in-

stallations in developed hospital reference buildings as a new approach to cost-fitness assessment [26], an analysis of metal oxide nanofluids for flat-plate solar collectors [27], and energy saving and emission reduction for industrial motor rewinding and replacement [28]. Additionally, some scholars have offered novel alternatives such as zero net energy building improvement guidelines for office buildings [29], a method of linking residential electricity usage and tropical outdoor climate [30], and energy efficiency packages for tenant installations undergoing laboratory testing and verification of indoor energy efficiency and environmental quality [31].

### 1.1.5. Measures and Assessments of Energy Saving

Measures and assessments have also been investigated. In the study of [32], various energy-saving measures were implemented. However, the use of variable speed drives has been found to be economical for various energy-saving measures, using larger motors for higher speed reductions and offering significant reductions in emissions. In other studies, energy saving has been measured by other methods such as assessing the energy flexibility of a building group under various regulatory factors [33], environmental, life cycle, economic and energy optimization assessment for building renovation [34], and an analysis of key investment policies, service reliability and social sustainability through access to electricity [35].

### 1.2. Prior Research on Condominiums

In recent decades, myriad interventions, particularly condominiums, have been proposed. The interventions are mainly concerned with cooling and heating systems, technology, building operation and energy options.

### 1.2.1. Cooling and Heating Systems

Some scholars [36,37] have proposed cooling systems for energy efficiency. For instance, one study [38] proposed a model for estimating cooling energy demands at the beginning of the condominium design. Condominium annual cooling energy demands can be reduced when important design variables that can be decided at an early stage for energy efficient design are taken into account. Similarly, numerous studies [39–46] examined the systems for energy efficiency. For instance, one study [47] built a model to estimate the final energy intensity of the building sector for space heating, household hot water, electricity for space cooling, and electricity for spaceless cooling in New York City, this model assumed that the final use depends largely on the functionality of the building, not on the type of construction or age of the building. Lastly, a study [48] examined the impact of conventional air conditioning design and installation issues, estimating that 31% of the systems were oversized, leading to excessive peak demand of 41 MW. Replacing larger systems with more accurate units has the potential for savings of 81 MW and enabling an average system saving of 18% of energy and 20% by sealing pipe leaks and servicing air-conditioning, respectively.

Other scholars have proposed heating systems for energy efficiency. For instance, one study [44] proposed a novel approach to accurately calculate energy and economic savings through heat meters and temperature control valves in residential buildings. In addition, a study [49] suggested that residential smart thermostat use can improve energy savings. Lastly, one study [39] offered a new HVAC system for solar thermal condominiums and cold weather outside by combining various elements in condominiums. Similarly, [36] found the significance of the residents' behavior in heat adaptation in conjunction with indoor air temperature measurements in condominiums equipped with home energy management systems. Some studies, like [42], examined the implementation of ISO 50001 to reduce the heat load for a social building stock. The results confirmed the effectiveness of ISO 50001 as an enterprise tool for managing energy in a systematic way.

1.2.2. Technology and Building Operation and Renovation

Several studies have attempted various technological approaches for specific purposes, like a retrofit approach for the net zero energy target of a four-floored office building in excellent weather to mitigate energy usage [47], the application of household energy saving options (HESO) to guide behavioral strategies of energy conservation [41], smart meter data clustering which yields a highly detailed and high-resolution electrical load dataset to develop a residential building's electricity consumption profile [9], simulation technology scenarios for converting condominiums into energy communities [37], and a multidimensional hybrid approach combining multiple interactions between observation-based and simulation-based data using the energy model's graphical software interface [1].

Other studies have paid attention to building operation and renovation. For instance, one study [49] examined the relationship between building characteristics and energy use. A wide range of energy intensities are unable to determine the trend of specific energy use with respect to differences in building operations, suggesting that many buildings can realize better energy efficiency by changing operating procedures. The building characteristics most associated with energy consumption are the fenestration ratio and boiler efficiency. Similarly, [49] suggested that the role that OSS clearly plays in making improvements can be instrumental in addressing many barriers preventing homeowners from renovating. OSS may be well-positioned in the future to address energy poverty by helping to access funding and involvement of real estate owners in renovating.

*1.3. Prior Research on Performance Evaluation of Energy Saving*

Several scholars have attempted to find effective ways to tackle energy saving and conduct a performance evaluation of various appliances for office and residential buildings. For example, one study [50] adopted advanced audio processing techniques to reduce the impacts of ambient noise. This experiment demonstrated that the proposed audio processing with the background noise cancellation algorithm improved the accuracy of room utilization estimates by approximately 11–12%, resulting in average ventilation power reduction of 3.54% compared to the absence of background noise cancellation. This proposed audio processing technique is likely to achieve a non-intrusive, cost-effective, robust and accurate solution for building occupancy estimates.

Additionally, a study [51] deployed the design and implementation of low-cost, wireless, and incremental sensor systems. This pilot project was able to (1) identify significant opportunities for energy savings due to idle intervals, (2) show an estimated battery lifetime of more than 5 years while accurately detecting occupancy, and (3) show that this system can potentially save energy from 10% to 15%.

Furthermore, a study [52] reviewed commonly available systems used in commercial office buildings for occupancy detection and control applications driven by demand and experimental results from evaluating the effectiveness of indoor chair sensors. This study provided detailed access information for demand-driven control applications.

Furthermore, a study [53] used WinOSS in a 1500 m$^2$ built environment for four weeks to determine its efficacy. WinOSS was chosen because it can provide detailed access information by leveraging the existing commodity WiFi infrastructure with WiFi-enabled mobile devices operated by occupants to outperform existing access detection techniques. Extensive results showed that WinOSS could provide detailed ownership information (including possession detection, counting and tracking) in an accurate, reliable, cost-effective and non-intrusive approach.

Lastly, a study [54] proposed an access monitoring system built on non-intrusive sensors capable of detecting indoor temperature, humidity, $CO_2$ concentration, door status, light, sound and movement. The efficiency of each sensor in the occupancy estimate was assessed. The sensor data is communicated wirelessly and real-time processing using a back-propagation (BP) artificial neural network (ANN) algorithm. For fifteen consecutive days, the test results reported an overall detection rate of over 90%, indicating the proposed system's ability to monitor real-time multi-room access data to support demand-driven

HVAC operations. These prior studies have proposed performance evaluation for energy efficiency and cost-effectiveness in office and residential buildings.

Collectively, smart technologies, models, and approaches have been studied extensively. Despite their energy efficiency and cost-effectiveness, few studies have paid attention to existing electrical appliances. Currently, little is known about how to improve these appliances for better energy efficiency and cost-effectiveness management. It has been observed by numerous electrical engineering experts, like [45] and two of the experts in this study, that the sizes of the existing electrical appliances that have been currently installed in condominiums in cities like Bangkok, are too large for the levels of energy usage and, as a result, the operation of the appliances appears to be significantly lower than their optimal capacity. It is estimated that downsizing the currently installed and used electrical appliances can help condominiums achieve energy efficiency and cost effectiveness management. However, the estimation has never been investigated and this was a research gap that needed to be further investigated. This investigation would yield significant contributions at the national and global levels since high rising energy consumption is one the major causes of the global crises such as global warming and climate change. The world has called for energy consumption reduction. It was, therefore, essential to attempt this research gap for a more ecofriendly world.

Drawing upon the observation and assumption, this study aimed to validate the aforementioned assumption of the size and draw insights in order to provide a proposal for energy efficiency and cost-effectiveness management of condominiums in Bangkok, Thailand, and other countries which share similar features and contexts. To seek solutions to the rising energy demands in Bangkok and elsewhere, we therefore, formulated the following three research questions.

- What are the profiles of condominiums and caretakers' perspectives of electric appliances and energy consumptions?
- What are the experts' perspectives on actual energy consumption and size of electrical appliances in the condominiums?
- What proposal for energy efficiency and cost-effectiveness management of condominiums should be?

## 2. Materials and Methods

### 2.1. Model and Assumption

To answer the aforementioned research questions, the conceptual model of this study could be photographically diagrammed as seen in Figure 1.

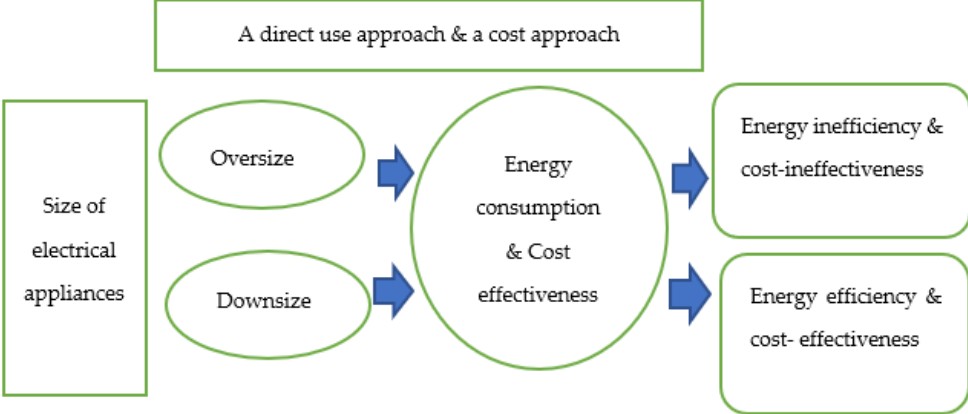

**Figure 1.** Impacts of electrical appliance sizes on energy consumption.

The model of the study above explains potential impacts of electrical appliance sizes on energy consumption. Sizes of electrical appliances have positive and negative impacts on energy consumption and cost-effectiveness. Oversized appliances consume more energy

and cost more than those fit for the actual energy usage of condominiums. Therefore, with respect to the direct use and cost approaches, the size of appliances plays a vital role in energy efficiency and cost-effectiveness management. Drawing upon the model above, the assumption of this study could be photographically diagrammed as seen in Figure 2.

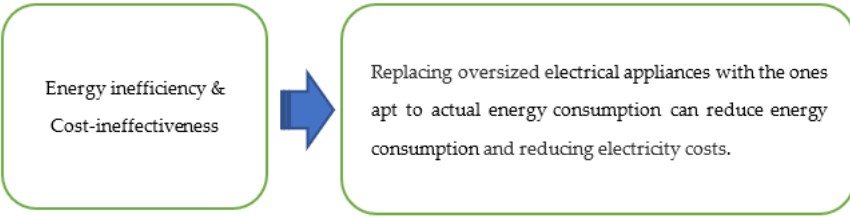

**Figure 2.** Management constraints and the assumption of the study.

*2.2. Design*

This study was based on a mixed-method research design. The population was 2944 condominiums in the central part of Bangkok, the capital of Thailand. The samples of 485 transformers were determined by Taro Yamane calculation. The data elicitation process followed these procedures. This study first surveyed the general profiles of condominiums and key facts (e.g., technical data) regarding energy assumptions of the condominiums. The survey data were collected 485 transformers from 20.00 p.m. to 23.00 p.m., which is the family time period, during March–April 2021, which is summer in Thailand. These periods were selected due to their highest electricity usage compared to other periods of the day and other seasons of the year. Thereafter, it moved on to the condominium caretakers' assessments of the suitability of the electrical appliances for the energy consumption. Lastly, in comparison with the caretakers' assessments, the assessments by electrical experts as key informants and in-depth interviews of their insights in regards to the assessments were conducted and compared with the key facts.

Expert opinion, a technique used in scientific research in various fields which guarantee the validity of the methodology to some extent, was purposefully chosen because it was used as a predictive tool with different types of solutions such as in the study of [55], which used this method to identify important and difficult concepts by experts, and in the study of [56], which used expert opinion to facilitate their investigation in professional contexts. The main objective of this study was to obtain concise answers to the predictive assumption of oversized appliances before moving to a more robust stage (namely implementation, experiment, and evaluation). This could be achieved in a timely manner with the aid of experts who share and exchange information, knowledge and experience.

*2.3. Participants*

Participants in this study fell into two groups: 7 electrical engineering experts and 15 heads of condominium technician caretakers. As key informants, all electrical engineering experts were purposefully selected because they met the criteria of this study, including qualification, experience, and expertise. On qualification, all of the experts received at least master's degrees in electrical engineering and half of them received doctoral degrees in electrical engineering. On expertise, all have high expertise in electrical systems design, energy saving, electrical protection, electric motor control, electrical design and installation, techniques for improving power factor, lighting systems, lighting system design, optimal DG placement in a smart distribution grid, and so on. On experience, all are university lecturers in the electrical engineering department and half of them are chairs of doctoral programs in electrical engineering. Additionally, all hold academic titles and have produced an average of 50 publications.

More importantly, all are members of the council of engineers. Some are committee members of the council of engineers and one of them was the president of electrical engineering department of the Engineering Institute of Thailand. In addition, they all have

been working in this field between 15 and 40 years in high-ranking positions in the public and private sectors, some of which include: board of professional knowledge of testing engineer qualification and ordinary engineer, electrical engineering, council of engineers; former director of district metropolitan electricity authority; former director of the bureau of structural engineering and systems engineering; former director of the systems department of public works and town and country planning; digital assistant governor provincial electricity authority (PEA); district director governor metropolitan electricity authority (MEA); chairman of the board of directors and chairman of the managing director, SECCO Engineering Company Limited, and director of the district electricity metropolitan electricity authority.

These key informants were interviewed by two researchers and one research assistant who was well-trained by the researchers to perform in response to the objectives of this study. All interviewers hold a doctoral degree or candidate status in electrical engineering, teaching positions in department of electrical engineering and related fields. All are members of the council of engineers and one of these interviewers is an IEEE member.

*2.4. Instruments for Data Elicitation*

The instruments for data elicitation included a questionnaire, a record of technical data, an assessment by condominium caretakers, an assessment by experts, and an in-depth interview respectively.

2.4.1. A Questionnaire

This instrument aimed to survey the general background of condominiums, areas and important features to understand the profiles of the condominiums in this study. The features included the locations, the height of buildings, the sizes of rooms, the number of occupants, the numbers of residents in one room, the energy consumption rates, and the types of distribution transformers installed in the buildings. In addition, the technical facts of the energy consumption of the buildings in this study were also recorded to draw key data which provided basis for comparison and assessment by condominium technical caretakers and the electrical engineering experts, including the average numbers of floors and the average transformer size.

2.4.2. A Record of Technical Data

This instrument aimed to draw technical facts of the average current ratings, the actual electricity consumption of a building and the average transformer size with the average current ratings, the average actual electricity consumption in comparison with the rated current of the transformer, the average actual current consumption within the utilization level of the current transformer rating that can be supplied, the average actual amounts of current in comparison with the ampere trip of circuit breaker, provisions of dry-type transformers used in condominiums equipped with fans, the average transformer size of supply in comparison with normal rated current, the supply of an average instantaneous overcurrent, the average actual current of the rated current of the transformer increased by the cooling fan, the average of C.B.'s AT and the average of C.B.'s AF, and the average of active capacitor. These key technical facts were recorded to provide the basis for the results of the assessment by condominium technical caretakers and by the electrical engineering experts.

2.4.3. An Assessment by Condominium Technical Caretakers

This assessment by condominium technical caretakers of electric appliances and energy consumptions, focusing on the suitability of key items (namely the settings of the main circuit breaker for the AF size, the AT setting of the main circuit breaker for the level of electricity usage, the use of the central current for the AF size and so on). The assessment fell into three levels, namely suitable, not sure, and unsuitable. The results of this assessment were compared with the recorded facts, and those of the electrical engineering experts' assessment through in-depth interviews.

### 2.4.4. An Assessment by Electrical Engineering Experts

This assessment by electrical engineering experts, focusing on the suitability of key issues (namely the suitability of current electrical appliances for AF of the main C.B., the suitability of current electrical appliances for AT of the main C.B., the suitability of the overall current consumption for the size of the transformer and so on). The assessment fell into the levels, namely agree, neutral, and disagree.

### 2.4.5. In-Depth Interviews

In-depth interviews aimed to draw insights from the experts' perspectives on the related issues. The interviews consisted of two parts. One aimed to assess the suitability of the electrical appliances for the actual usages to gain more insightful explanation and details of the assessments. The other aimed to assess the suitability of the new sizes of electrical appliances proposed for energy efficiency management, particularly in terms of energy consumption.

### *2.5. Data Analysis*

The quantitative data drawn from the questionnaire, the record of technical data, and the two assessments were analyzed by descriptive statistics, e.g., mean, standard deviation, and percentage. The qualitative were data drawn from the two in-depth interviews. The data from all instruments and were quantitatively and qualitatively analyzed to draw the conclusions of the study.

### *2.6. Reliability and Validity Check*

The questionnaire was constructed in response to the research questions and then sent to five reviewers for the content validity check and the item-objective congruence (IOC) evaluation. The reliability check of the questionnaire was tested by the Alpha Cronbach's coefficient test. The value was 0.76, indicating moderate value. The two assessments of the condominium technical caretakers and the electrical engineering experts followed the same procedures as the questionnaires. The values were 0.78 and 0.90, indicating moderate and high values, respectively. The experts' perspectives were coded, recoded, compared, and analyzed. The conclusion of this study was drawn from triangulation of these various data sources.

## 3. Results

### *3.1. Results of the Study*

### 3.1.1. Profiles of Condominiums in Bangkok

The profiles of 485 condominiums in this study (namely locations, heights, sizes of room, numbers of residents, numbers of occupants, and distribution transformers) could be graphically summarized in percentages as seen in Figure 3.

Figure 3 illustrates proportions of locations, heights, room sizes, numbers of residents, numbers of occupants, and distribution transformers of the condominiums in this study. On location, the majority of condominiums (85%) were located adjacent to the main road (29%), in alleys (29%), near the main road (29%), and near BTS line (27.6%) respectively. The minority of condominiums (12.7%) ere located near the MRT line. On height, the majority of condominiums (85.5%) were of 23–50 m high (30.5%), more than 51–100 m high (27.5%) and more than 100 m (27.5%), respectively. The minority of condominiums were less than or equal to 23 m high (14.4%). On sizes of room, the sizes were arranged from the largest to the smallest as follows: 31–40 square metres (38.1%), more than 50 square metres (25.6%), 20–30 square metres (20.8%), and 41–50 square metres (15.5%), respectively. On numbers of residents, the majority of rooms (74.9%) had 2 residents, while the minority (18.6%) had 3 residents and the rest (6.5%) had either more than 3 residents or just one. On numbers of occupants, the proportion of occupants can be arranged as follows: 61–80% (48.2%), 41–60% (25.6%), 81–100% (18.5%), and 20–40% (7.7%), respectively. Lastly, on distribution

transformers, the transformer in conjunction with the room accounted for 82.6% while the central distribution transformer accounted for 17.4%.

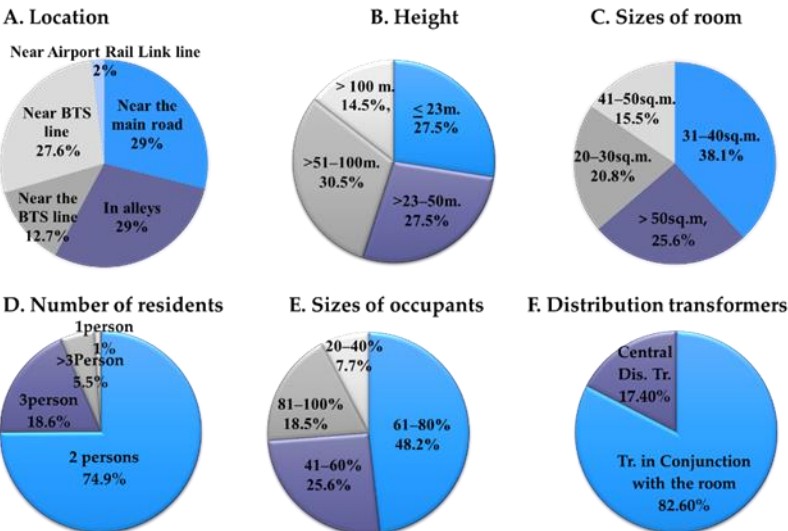

**Figure 3.** Percentages of locations, heights, sizes of room, number of residents, numbers of occupants, and distribution transformers.

### 3.1.2. Technical Data of Electrical Appliances

Technical data recorded by the building caretaker involved the total electric power consumption from 20.00 p.m. to 23.00 p.m. every day, as it was the period of highest energy consumption. It should also be noted that the data were collected during the COVID lockdown, resulting in higher energy consumption than normal conditions due to work from home (WFH) and online learning.

The details of the data are as follows. The average numbers of floors was 23.81. The actual electricity consumption of a building was 145.65 A while the average transformer size is 1432 kVA with the average current ratings of 2034 A. This indicated that the average actual electricity consumption of the building (145.65 A) was equal to 7.2% of the 2034 A average current. The average actual electricity consumption equaled 10.9% of the rated current of the transformer, which falls within the utilization level of 0–10% of the current transformer rating that can be supplied. The average actual amounts of current equaled only 6.4% of the ampere trip of circuit breaker. Dry-type transformers used in condominiums equipped with fans can provide up to 40% over current usage. The average transformer size of 1431 kVA can supply a normal rated current of 2034 A and can supply an average instantaneous overcurrent of 2848 A. The average actual current is 7.8% of the rated current of the transformer increased by the cooling fan. The average of C.B.'s AT was 2548 A and the average of C.B.'s AF was 2667 A. Lastly, the average of active capacitor was 0.13 step, indicating that the capacitor bank was used less than 1 step while the total capacitor equals to 10.48 steps.

### 3.1.3. An Assessment by Condominium Technical Caretakers

Table 1 shows mean scores and standard deviation of condominium technical caretakers' assessments of the suitability of electrical appliances for the condominium electrical systems.

**Table 1.** Mean and standard deviation of caretakers' assessments of the suitability of electrical appliances.

| Questions<br>Do You Think... ? | Mean | S.D. |
|---|---|---|
| 1.  the settings of the main circuit breaker are suitable for the AF size | 0.95 | 0.28 |
| 2.  the AT setting of the main circuit breaker is suitable for the level of electricity usage | 0.95 | 0.28 |
| 3.  the use of the central current is suitable for the AF size | 0.92 | 0.30 |
| 4.  the AT setting at the central C.B. suitable for the use of central electricity | 0.96 | 0.21 |
| 5.  the level of electricity usage is suitable for Tie C.B.'s AF size | 0.92 | 0.27 |
| 6.  the AT setting at Tie C.B. is suitable for the use of electricity | 0.90 | 0.32 |
| 7.  the overall electricity usage is suitable for the size of the transformer | 0.93 | 0.25 |
| 8.  the steps used are suitable for the total number of the capacitor bank's steps | 0.89 | 0.35 |
| 9.  the design of the transformer and the C.B. that is much larger than the actual overall current power consumption is suitable for the level of electrical system | 0.83 | 0.49 |
| Total | 0.55 | 0.76 |

All caretakers agreed with the suitability of the electrical appliances for the capacity of the condominiums. The results of the assessment can be arranged as follows: the AT setting of the central C.B. is suitable for the use of central electricity (=0.96, S.D. = 0.21, the settings of the main circuit breaker are suitable for the AF size (=0.95, S.D. = 0.28, the AT setting of the main circuit breaker is suitable for the level of electricity usage (=0.95, S.D. = 0.28), the overall electricity usage is suitable for the size of the transformer (=0.93, S.D. = 0.25), the use of the central current is suitable for the AF size (=0.92, S.D. = 0.3), the level of electricity usage is suitable for Tie C.B's AF size (=0.92, S.D. = 0.27), the AT setting at Tie C.B. is suitable for the use of electricity (=0.90, S.D. = 0.32), the steps used are suitable for the total number of the capacitor bank's steps (=0.89, S.D. = 0.35), the design of the transformer and the C.B that is much larger than the actual overall current power consumption is suitable for the level of electrical system (=0.83, S.D. = 0.49) and the selection of electrical equipment that is too large for usage suitable for the electrical system (=0.55, S.D. = 0.76), respectively. This indicated that condominium technical caretakers agreed with the suitability of electrical appliances for the condominium electrical systems. It should be noted that the condominium technical caretakers are stakeholders. The results of their assessment needed to be verified by the third party highly qualified by their experience and expertise, as shown in the results of the next research questions.

### 3.2. Results of the Expert

The experts' perspectives on actual energy consumption and sizes of electrical appliances in the condominiums are illustrated in Table 2.

All seven experts unanimously came to a consensus on the unsuitability of electrical appliances for the condominium electrical systems in all items.

The results of the aforementioned assessment are supported by in-depth interviews. One expert revealed that the usage load of condominiums is generally excessive according to the standard criteria, while one expert pinpointed that the installers and users lacked knowledge of the selection of transformer protection devices. Another expert suggested that the building's power consumption encompasses various variables (namely residential, business, and corporate marketing factors). These factors need to be taken into consideration. Similarly, one expert confirmed that the selection and sizes of electrical equipment should take several factors into account (namely appropriateness for the needs of electricity, the economics principle of and safety).

**Table 2.** Percentage of experts' assessments of suitability of electrical appliances for the condominium electrical systems.

| Suitability of Electrical Appliances for the Electrical Systems | Level of Agreement (%) | | |
|---|---|---|---|
| | Agree | Neutral | Disagree |
| 1. Are the current electrical appliances suitable for AF of the main C.B? | 0 | 0 | 100 |
| 2. Are the current electrical appliances suitable for AT of the main C.B? | 0 | 0 | 100 |
| 3. Is the central current suitable for AT of C.B.? | 0 | 0 | 100 |
| 4. Is the central current suitable for AF of C.B.? | 0 | 0 | 100 |
| 5. Is the current suitable for AT of Tie C.B.? | 0 | 0 | 100 |
| 6. Is the current suitable for AF of Tie C.B.? | 0 | 0 | 100 |
| 7. Is the overall current consumption suitable for the size of the transformer? | 0 | 0 | 100 |
| 8. Is the total active step of capacitor bank suitable for operation? | 0 | 0 | 100 |
| 9. Is the design of Tr and C.B., which is larger than the actual current consumption, suitable for electrical systems? | 0 | 0 | 100 |
| 10. Are various electrical appliances, which are too large for the actual usage, suitable for electrical systems? | 0 | 0 | 100 |

It is also essential to verify the experts' assessments. One of the most effective ways to do so is to examine the facts graphically, as shown in the bar chart below.

Figure 4 compares the ratio between the actual usage and the capacity of electrical appliances. This indicated crucial differences between the actual usage and the capacity of the appliances. The ratios of the actual usage are significantly lower than the capacity of the appliances as summarized Table 3, below.

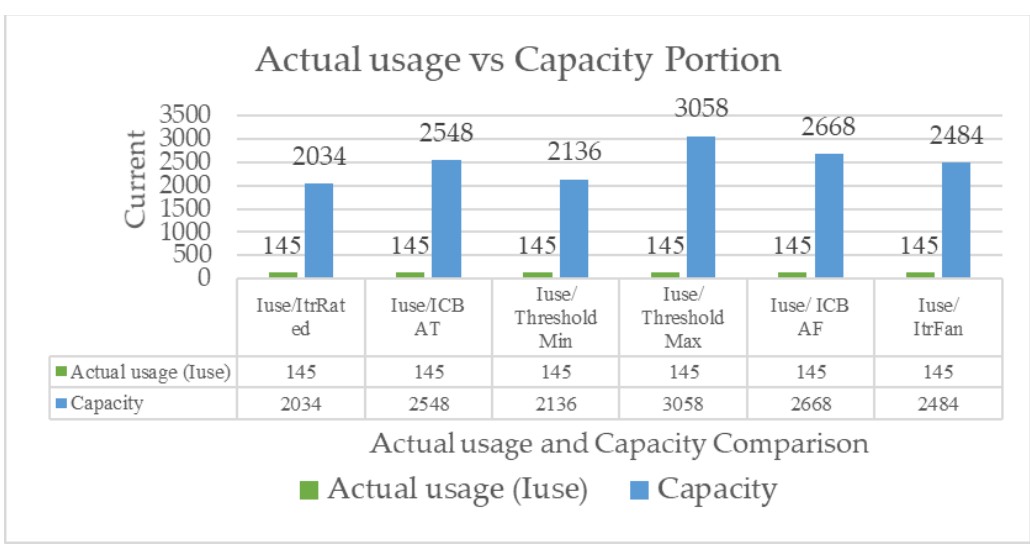

**Figure 4.** Comparison of ratios between actual usage and capacity of electrical appliances.

**Table 3.** Comparison of ratios between actual usage and capacity of electrical appliances.

| Item of Comparison | Actual Usage (Iuse) | Capacity | Difference | Percentage |
|---|---|---|---|---|
| Iuse/ItrRated | 145 | 2034 | 1889 | 7.1% |
| Iuse/ICB AT | 145 | 2548 | 2403 | 5.7% |
| Iuse/ICB AF | 145 | 2668 | 2523 | 5.4% |
| Iuse/ItrFan | 145 | 2484 | 2339 | 5.8% |

Table 3 compares the ratios between the actual usage and the capacity of electrical appliances. There are wide differences between the actual usage and capacity of all compared items, from the most to the least, as follows: Iuse/ICB AF (2668), Iuse/ICB AT (2548), Iuse/ItrFan (2484), and Iuse/ItrRated (Iuse/ItrRated) respectively. In terms of percentages, the actual usage was between 5.4% and 7.1% of the capacity. This indicated that the actual usage was much lower than the capacity of the electrical appliances installed in the condominiums.

It is clearly noted that the experts' consensus was consistent with the facts in Figures 4 and 5 regarding the unsuitability of appliances, and almost absolutely contradict the condominium technician caretakers. It could be concluded that the electrical appliances installed in the condominiums were unsuitable for the actual usage and this may be the key factor of high energy consumption. It is essential to find solutions to energy efficiency and cost effectiveness management, as proposed in the next part.

**Figure 5.** Single line diagram displaying utility of power supply.

*3.3. Proposal for Energy Efficiency and Cost Effectiveness Management of Condominiums*

Drawing upon the former research questions, we therefore developed two proposals: (a) managing utility of power supply and (b) downsizing the current electrical appliances installed in the condominiums, as seen in the details below.

Figure 5 illustrates the utility of power supply in a single line diagram. The power transformer rating is 1600 kVA while the average actual current consumption is 146 A, which is only 6.3% of the power supply rating of a 1600 kVA transformer. Therefore, for consumption management, a building which has 2 transformers or more for electrical reliability and preparation for the case of malfunction can either rely on only one transformer or employ both transformers by planning to manage them alternately one after another.

A. Managing utility of power supply.
B. Downsizing the current electrical appliances.

Table 4 compares two sizes of electrical appliances (namely current standard sizes and new or downsized alternatives) which will yield positive impacts on cost effectiveness management. If a condominium chooses the standard sizes of the items above, which are currently used, (namely MV Switchgear 2 sets, Dry Type transformer 2 sets 80,000, LV cable 10 m. (XLPE), main distribution board, Busduct (MDB-DB), generator (20% of Tr.), generator installation), the total cost is $565,333. On the other hand, if it chooses the new sizes which have been proposed in this study, the total cost is only $305,333, a cost reduction of $260,000 or 54%. This indicated that new appliances can lead to cost effectiveness successfully.

**Table 4.** Comparison of two sizes of electrical appliances for cost effectiveness management.

| No | Electrical Appliances | Current Standard Sizes | | Downsized Alternatives | | Diff. |
|---|---|---|---|---|---|---|
| | | Size | Cost (USD) | Size | Cost (USD) | (USD) |
| 1 | MV Switchgear 2 Ea | 200 A. 24 kV. | 50,000 | 200 A. 24 kV. | 50,000 | 0.00 |
| 2 | Transformer 2 Ea | 2 × 1600 | 146,667 | 2 × 630 | 66,667 | 80,000 |
| 3 | LV Cable 10 m. (XLPE) | 2 × [6 × (3 × 240/185 N)] | 15,333 | 2 × [3 × (3 × 240/185 N)] | 5333 | 10,000 |
| 4 | MDB | 2500 A. | 83,333 | 1000 A. | 46,667 | 36,667 |
| 5 | Busduct (MDB-DB) | 2 × 2000 A. | 183,333 | 2 × 1000 A. | 93,333 | 90,000 |
| 6 | Generator (20% of Tr.) | 350 kVA | 70,000 | 130 kVA | 33,333 | 36,667 |
| 7 | Generator Installation | 350 kVA | 16,667 | 130 kVA | 10,000 | 6667 |
| | Total Cost | | 565,333 | Reduced 54% | 305,333 | 260,000 |

In addition, in terms of energy efficiency, the in-depth interviews of the electrical engineering experts reviewed that they all agreed with the new proposal for these reasons. Reducing sizes as offered here will yield more benefits in terms of safety because, if the electrical appliance is substantially oversized, the circuit breaker can cause malfunction in case of overloading. In terms of installation, large electrical appliances also waste space. In terms of maintenance, this demands considerably higher costs. In term of energy efficiency, there is enormous power loss when more than one oversized appliance operates simultaneously, and because all transformers supply less than 10% of their maximum capacity and the transformers will never operate at their maximum capacity. This indicated potential energy inefficiency.

## 4. Discussion

The findings of this study lend support to the conclusions of several studies in the major issues discussed below.

Like the study of [47] indicating that the energy intensity of the building sector depends on the functionality of the building, not on construction or age of the building, this present study found that the electrical functionality of the building determines the energy consumption and intensity of the building (namely MV switchgear 2 sets, dry type transformer 2 sets 80,000, LV cable 10 m. (XLPE), main distribution board, busduct (MDB-DB), generator (20% of Tr.), and generator installation).

Similarly, like the study of [38] which concluded that estimating the cooling energy demand at the beginning of the condominium design helps reduce the energy consumption of the building, this present study proposed that estimating appropriate sizes of electrical appliances at an early stage can help mitigate the energy usage of the condominiums.

This present study suggests downsizing these electrical appliances for energy efficiency because the actual usage lies only at 5.4% to 7.1% of the capacity. The finding of this study is consistent with that of [48], which estimated that 31% of the systems were

oversized and led to excessive peak demand of 41 MW. This study therefore suggests that replacing larger systems with more accurate units appropriate to the optimal capacity has the potential for savings of 81 MW and enables an average system saving of 18% of energy and 20% by sealing pipe leaks and servicing air-conditioning. However, unlike the study of [48], this present study found that the financial value indicates cost-effectiveness, i.e., 54% reduction of the total cost.

The suggestions of this study which is an attempt to dealt with energy saving lends support to those of prior research such as [51], which deployed the design and implementation of low-cost, wireless, and incremental sensor systems and found that this system could potentially save energy from 10% to 15% [50], which demonstrated that the proposed audio processing with the background noise cancellation algorithm improved the accuracy of room utilization estimates by approximately 11–12%, resulting in average ventilation power of 3.54% reduction compared to the absence of background noise cancellation [53], which showed that WinOSS, which could provide detailed access information by leveraging the existing commodity WiFi infrastructure with WiFi-enabled mobile devices operated by occupants outperforms existing access detection techniques and could provide detailed ownership information (including possession detection, counting and tracking) [54], which reported an overall detection rate of over 90%, indicating the proposed system's ability to monitor real-time multi-room access data to support demand-driven HVAC operations; and [52], which evaluated the effectiveness of indoor chair sensors and provided detailed access information for demand-driven control applications. Like the suggestions of those studies, this study provides useful details of appliances to be downsized, such as MV Switchgear 2 sets, dry type transformer 2 sets 80,000, LV cable 10 m. (XLPE), and so on, which is assumed to yield a 54% cost reduction. However, unlike those studies, the suggestions are based on expert knowledge, meaning further implementation, experimentation, and performance evaluations of the proposed appliances with solid and extensive results is required evidence to provide validation of the assumption and for accuracy.

Although the assumption of 54% reduction by downsizing these existing appliances in the condominiums in coupled with other benefits such as cost reduction as proposed in this study still needs experimentally examined, the results of this study provide some insights and practical implications for researchers to further validate the assumption, which might result in higher or lower numbers than the 54% reduction calciated in this study. This variation of reduction depends on factors relating to the building (such as the age of the building, the structure of the building, and other features). This assumed reduction rate, more or less, will shed some light into how all condominiums could consider the ways to save energy as proposed in this study. The higher the number of condominiums to consider this energy-saving proposal, the better the environment in Bangkok will be. If this proposal diffuses to other cities, this will contribute to the national environment as a whole. If this proposal diffuses worldwide, this will mitigate the impacts of the global crises such as global warming and climate change. This mitigation will heal the world and make the world a better place than it actually is now. It could be concluded that this study will create a more ecofriendly environment which will ultimately contribute to all mankind.

## 5. Conclusions

Three conclusions could be drawn from the aforementioned results of the study based on the research questions.

First, on the profiles of condominiums and caretakers' perspectives of electric appliances and energy consumptions, the condominium technical caretakers agreed with the suitability of electrical appliances for the condominium electrical systems. Their opinions were inconsistent with the factual records of energy consumption.

Second, on the experts' perspectives on actual energy consumption and size of electrical appliances in the condominiums, all experts unanimously agreed on the unsuitability of the electrical appliances for the condominium electrical systems.

Lastly, data records reveal that most condominiums installed oversized electric appliances and the actual energy usage lies between 5.4% and 7.1% of the capacity. This study proposed to downsize key appliances, namely MV Switchgear 2 sets, dry type transformer 2 sets 80,000, LV cable 10 m. (XLPE), main distribution board, Busduct (MDB-DB), generator (20% of Tr.), generator installation, resulting in 54% of cost reduction.

## 6. Limitation of the Study

The major limitation of this study lies in the methodology of the study. Since this study mainly elicited data from expert opinion, knowledge, and expertise, the data might have predictive values but the prediction weights less than evidence drawn from the extensive results of scientific investigation. This study needs further investigations.

## 7. Future Inquiry

As this study aimed to verify the assumptions by experts' perspectives in comparison with factual data record within a short period of time, it is only an initial step which needs further inquiry for more solid data for confirmed validation. Future inquiry should be direct towards a pilot project of several short case studies where the proposed appliances would be installed in condominium buildings and extensive experiments would be conducted to validate the performance and evaluate the accuracy. In the long term, experiments of longitudinal studies to validate the accuracy of the implementation should be attempted. In addition, factors like the structure and age of the buildings which can affect the installed appliances as proposed by this study, should be taken into account to ensure the highest level of energy efficiency and cost-effectiveness.

**Author Contributions:** Conceptualization, T.B. and K.W.; data curation, T.B.; formal analysis, T.B. and K.W.; funding acquisition, T.B.; project administration, T.B. and K.W.; software analysis, T.B. and K.W.; supervision, T.B.; writing—original draft preparation, T.B. and K.W.; and writing—review and editing, T.B. and K.W. All authors have read and agreed to the published version of the manuscript.

**Funding:** This research was financially supported by Mahasarakham University, Thailand.

**Institutional Review Board Statement:** Not applicable.

**Informed Consent Statement:** Not applicable.

**Data Availability Statement:** Not applicable.

**Acknowledgments:** This research was financially supported by Mahasarakham Business School, Mahasarakham University, Thailand. Also, the authors would like to thank Worawat Sa-ngiamvibool, Thailand, for data analysis and help in the preparation of the original draft for this work. Finally, the authors also owe to Intisarn Chaiyasuk for his English language consulting time and proofreading the whole paper.

**Conflicts of Interest:** The authors declare no conflict of interest.

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
