# Peer review of "Oversized Electrical Appliance Impacts on Condominium Energy Efficiency and Cost-Effectiveness Management: Experts’ Perspectives"

_asi, doi:10.3390/asi4040098_

Round 1

Reviewer 1 Report

The article concerns very interesting analyzes related to energy efficiency management in an apartment block. The economic analysis was performed by using expert surveys that showed oversizing of many energy devices. After their identification (with the participation of electrical engineering experts), it was proposed to replace oversized electrical devices with new ones with a reduced demand for electrical power. This cut costs by up to half. The conducted analyzes showed that the energy consumption of buildings depends mainly on their functionality, and to a much lesser extent on the structure or age of the building. The conducted analyzes have also shown that the selection of electrical devices for apartment blocks is wrong and the power reduction of some of them leads to very large financial savings. The article is written at a good substantive level and is perfectly suitable for publication, but the overall tone of the work is not very scientific. Since the analysis leads to practical guidelines related to the electrical installations of apartment blocks, I would direct my work to the Applied Sciences journal rather than Energies. However, I leave this decision to the editors. 

Author Response

Thanks for the advice from the reviewer. However, this paper integrated knowledge between social sciences and science to obtain new knowledge. In addition, the increasing number of students interested in studying business administration and digital innovation programs. Therefore, there may be a large number of integrated research increases. If publishers can open up more opportunities for articles in this field, it will be a good opportunity.

Reviewer 2 Report

It is a detailed work of analyzing the effects of oversize electrical appliances to the environment, especially for the energy efficiency.

  1. In addition to down sizing the electrical appliances, other similar ideas such as occupance-based operation or demand-based operation might need to be mentioned in this revised work, too.

[1] Q. Huang, "Occupancy-driven energy efficient buildings using audio processing with background sound cancellation", Buildings, vol. 8, no. 6, pp. 1-16, 2018.

[2] Agarwal, Y.; Balaji, B.; Gupta, R.; Lyles, J.; Wei, M.; Weng, T. Occupancy-driven energy management for smart building automation. In Proceedings of the ACM Workshop on Embedded Sensing Systems for Energy-Efficiency in Building, Zurich, Switzerland, 3–5 November 2010; pp. 1–6.

[3] Labeodan, T.; Zeiler, W.; Boxem, G.; Zhao, Y. Occupancy measurement in commercial office buildings for demand-driven control applications-a survey and detection system evaluation. Energy Build. 2015, 93, 303–314.

[4] Zou, H.; Jiang, H.; Yang, J.; Xie, L.; Spanos, C. Non-Intrusive Occupancy Sensing in Commercial Buildings. Energy Build. 2017, 154, 633–643.

[5] Yang, Z.; Li, N.; Gerber, B. A non-intrusive occupancy monitoring system for demand driven HVAC operations. In Proceedings of the Construction Research Congress, West Lafayette, IN, USA, 21–23 May 2012; pp. 828–837.

2. In Section 4, please also compare your work with these above references to show that the benefits of downsizing and occupance-adaptive control

Author Response

Prior research on performance evaluation of energy saving

Several scholars have attempted to find effective ways to tackle with energy saving and conduct a performance evaluation of various appliances for office and residential buildings. For example, [51] adopt advanced audio-processing techniques to reduce the impacts of ambient noise. This experiment demonstrated that the proposed audio processing with the background noise cancellation algorithm improved the accuracy of room utilization estimates by approximately 11–12%, resulting in average ventilation power of 3.54% reduction compared to the absence of background noise cancellation. This proposed audio-processing technique is likely to achieve a non-intrusive, cost-effective, robust and accurate solution for building occupancy estimates.

Additionally, [52] deployed the design and implementation of low-cost, wireless, and incremental sensor systems. This pilot project was able to (1) identify significant opportunities for energy savings due to idle intervals, (2) show an estimated battery lifetime of more than 5 years while accurately detecting occupancy, and (3) show that this system can potentially save energy from 10% to 15%.

Also, [53] reviewed commonly available systems used in commercial office buildings for occupancy detection and control applications driven by demand and experimental results from evaluating the effectiveness of indoor chair sensors. This study provided detailed access information for demand-driven control applications.

Forthermore, [54] used WinOSS in a 1500 m2 built environment for four weeks to determine its efficacy. WinOSS was chosen because it can provide detailed access information by leveraging the existing commodity WiFi infrastructure with WiFi-enabled mobile devices operated by occupants outperforms existing access detection techniques. Extensive results showed that WinOSS could provide detailed ownership information (including possession detection, counting and tracking) in an accurate, reliable, cost-effective and non-intrusive approach.

 Lastly, [55] proposed an access monitoring system built on non-intrusive sensors capable of detecting indoor temperature, humidity, CO2 concentration, door status, light, sound and movement. The efficiency of each sensor in the occupancy estimate is assessed. The sensor data is communicated wirelessly and real-time processing using a back-propagation (BP) artificial neural network (ANN) algorithm. For fifteen consecutive days, the test results reported an overall detection rate of over 90%, indicating the proposed system's ability to monitor real-time multi-room access data to support demand-driven HVAC operations. These prior studies have proposed performance evaluation for energy efficiency and cost-effectiveness in office and residential buildings.

The suggestions of this study which is an attempt to dealt with energy saving lends support to those of prio research such as [52], which deployed the design and implementation of low-cost, wireless, and incremental sensor systems and found that this system could potentially save energy from 10% to 15%, [51], which demonstrated that the proposed audio processing with the background noise cancellation algorithm improved the accuracy of room utilization estimates by approximately 11–12%, resulting in average ventilation power of 3.54% reduction compared to the absence of background noise cancellation, [54], which showed that WinOSS, which could provide detailed access information by leveraging the existing commodity WiFi infrastructure with WiFi-enabled mobile devices operated by occupants outperforms existing access detection techniques and could provide detailed ownership information (including possession detection, counting and tracking), [55], which reported an overall detection rate of over 90%, indicating the proposed system's ability to monitor real-time multi-room access data to support demand-driven HVAC operations, and [53], which evaluated the effectiveness of indoor chair sensors and provided detailed access information for demand-driven control applications. Like the suggestions of those studies, this study provides useful details of appliances to be downsized such as MV Switchgear 2 sets, Dry Type Transformer 2 sets 80,000, LV Cable 10m. (XLPE), and so on, which assumes to yield 54% cost reduction. However, unlike those studies, the suggestions are based on expert knowledge that still needs further implementation, experimentation, and performance evaluation of the proposed appliances with solid and extensive results as evidence to provide validation of the assumption and further for accuracy.

Reviewer 3 Report

The study is well designed and supported by a good number of sources. It would be good if the authors provided a clear indication of the research gap and also explained later in the discussion part how this gap was addressed and what are the general implications of the study to the wider society in Thailand and worldwide. The limitations of the study should be spelt out more clearly. The consistency of style across figures and tables is recommended. Adding more references to the methodology part could improve the soundness of the selected methods.

Some technical comments:

Line 44 has a duplication of words

The referencing style should be checked. For example, if the citation is in-text, does it need to include the name of the first author

Line 141 has an empty space

Line 178 requires referencing. Who are the experts and where did they mention the discussed issue

Check Figure 1 if the text in the figure is too big

Lines 295 and 304. Do these subheadings be more visible? This relates to all similar subheadings across the paper

Figure 3 needs to be rectified to be more representative

Author Response

Collectively, smart technologies, models, and approaches have been studied extensively. Despite of energy efficiency and cost-effectiveness, few studies have paid attention to existing electrical appliances that have been operating. Currently little is known about how to improve these appliances for better energy efficiency and cost-effectiveness management. It has been observed by numerous electrical engineering experts, like [45] and two of the experts in this study, that the sizes of the existing electrical appliances that have been currently installed in condominiums in cities, like in Bangkok, are too large for the levels of energy usage and, as a result, the operation of the appliances appears to be significantly lower than their optimal capacity. It is estimated that downsizing the currently installed and used electrical appliances can help condominiums achieve energy efficiency and cost effectiveness management. However, the estimation has never been investigated and this was a research gap that needed to be further investigated. This investigation would yield significant contributions to national and global levels since the high rising energy consumption is one the major causes of the global crises such as global warming and climate change. The world has called for energy consumption reduction. It was, therefore, essential to attempt this research gap for a more ecofriendly world.

Expert opinion, a technique used in scientific research in various fields, which guarantee the validity of the methodology to some extent, was purposefully chosen because it was used as a predictive tool with different types of solutions such as in the study of [57], which used this method to identify important and difficult concepts by experts, in the study of [56], which used expert pinion to facilitate their investigation in professional contexts and in the of [58], which used expert knowledge in risk assessment and decision analysis. The main objective of this study was to get concise answers to the predictive assumption of oversized appliances before moving to a more robust stage (namely implementation, experiment, and evaluation). This could be achieved in a timely manner with the aid of experts who share and exchange information, knowledge and experience.

              Although the assumption of 54% reduction by downsizing these existing appli-ances in the condominiums in coupled with other benefits such as cost reduction as proposed in this study still needs experimentally examined for validation of accuracy, the results of this study provide some insights and practical implications for research-ers to further validate the assumption, which might result in higher or lower numbers than 54% reduction as proposed in this study. This variation of reduction depends on factors relating to the building (such as the age of the building, the structure of the building, and other features). This assumed reduction rate, more or less, will shed some light to all condominiums to consider the way to save energy as proposed in this study. The higher numbers of condominiums consider this energy-saving proposal, the better the environment in Bangkok will be. If this proposal diffuses to other cities, this will contribute to the national environment as a whole. If this proposal diffuses worldwide, this will mitigate the impacts of the global crises such as global warming and climate change. This mitigation will heal the world and make the world a better place than it actually is now. It could be concluded that this study will create a more ecofriendly environment which will ultimately contribute to all mankind.

5.2. Limitation of the study

The major limitation of this study lies in the methodology of the study. Since this study mainly elicited data from expert opinion, knowledge, and expertise, the data might have predictive values but the prediction weights less than evidence drawn from extensive results of scientific investigation. This study needs further investigations.

5.3 Future inquiry

As this study aimed to verify the assumption by experts’ perspectives in comparison with factual data record within a short period of time, it is only an initial step which needs further inquiry for more solid data for confirmed validation. Future inquiry should direct towards a pilot project of several short case studies which will install the proposed appliances in condominium buildings and conduct extensive experiments to validate of the performance and evaluate the accuracy. In the long term, experiments of longitudinal studies to validate the accuracy of the implementation should be attempted. In addition, factors like the structure and age of the building, which can affect the installed appliances as proposed by this study, should be taken into account to ensure the highest level of energy efficiency and cost-effectiveness.

Round 2

Reviewer 3 Report

Thanks for the modifications